# Metabolomics: A Way Forward for Crop Improvement

**DOI:** 10.3390/metabo9120303

**Published:** 2019-12-14

**Authors:** Ali Razzaq, Bushra Sadia, Ali Raza, Muhammad Khalid Hameed, Fozia Saleem

**Affiliations:** 1Centre of Agricultural Biochemistry and Biotechnology (CABB), University of Agriculture, Faisalabad 38040, Pakistan; ali.razzaq254@gmail.com (A.R.); bushra.sadia@uaf.edu.pk (B.S.); 2Oil Crops Research Institute, Chinese Academy of Agricultural Sciences (CAAS), Wuhan 430062, China; alirazamughal143@gmail.com; 3School of Agriculture and Biology, Shanghai Jiao Tong University, Shanghai 200240, China; Khalid_khalid45@yahoo.com

**Keywords:** metabolomics, metabolic profiling, crop improvement, abiotic stress, biotic stress, mass spectrometry, metabolomics-assisted breeding

## Abstract

Metabolomics is an emerging branch of “omics” and it involves identification and quantification of metabolites and chemical footprints of cellular regulatory processes in different biological species. The metabolome is the total metabolite pool in an organism, which can be measured to characterize genetic or environmental variations. Metabolomics plays a significant role in exploring environment–gene interactions, mutant characterization, phenotyping, identification of biomarkers, and drug discovery. Metabolomics is a promising approach to decipher various metabolic networks that are linked with biotic and abiotic stress tolerance in plants. In this context, metabolomics-assisted breeding enables efficient screening for yield and stress tolerance of crops at the metabolic level. Advanced metabolomics analytical tools, like non-destructive nuclear magnetic resonance spectroscopy (NMR), liquid chromatography mass-spectroscopy (LC-MS), gas chromatography-mass spectrometry (GC-MS), high performance liquid chromatography (HPLC), and direct flow injection (DFI) mass spectrometry, have sped up metabolic profiling. Presently, integrating metabolomics with post-genomics tools has enabled efficient dissection of genetic and phenotypic association in crop plants. This review provides insight into the state-of-the-art plant metabolomics tools for crop improvement. Here, we describe the workflow of plant metabolomics research focusing on the elucidation of biotic and abiotic stress tolerance mechanisms in plants. Furthermore, the potential of metabolomics-assisted breeding for crop improvement and its future applications in speed breeding are also discussed. Mention has also been made of possible bottlenecks and future prospects of plant metabolomics.

## 1. Metabolomics: Significance in Plant Biology

Metabolomics is one of the emerging and fascinating approaches of omics tools, which has now been extensively applied for crop improvement. Metabolomics is crucial to studying the abiotic stress tolerance, pathogen resistance, robust ecotypes, and metabolic assisted-breeding of crops. To date, enormous progress has been made to develop modern metabolomics tools for crop improvement [1]. The plant kingdom contains a huge diversity of metabolites of approximately 200,000 compounds and the vast majority are still unknown. It is estimated that around 10000 secondary metabolites have been discovered in different plant species. The discovered metabolites are structurally different in their biochemical properties and functions and are considered very important in plant biology [2]. Modern metabolomics platforms are being exploited to explain complex biological pathways and explore hidden regulatory networks controlling crop growth and development. Metabolome consists of a complete set of low molecular weight metabolites within biological systems. Metabolomics research is primarily concerned with the identification and quantification of small molecules (<1500 Da), their chemical structure, and interactions within an organism [3]. Many metabolites are unique and play an important role in the crop yield and nutritional quality control [4]. Plant growth and development under different environmental conditions are greatly influenced by the synthesis of a large number of metabolites. Environmental metabolomics involves investigating the interactions of plants with their environment. It is based on a detailed evaluation of metabolite levels under a specific plant ecology to pinpoint the effects on plant adaptation and any modifications in their genetic architecture. It provides a platform to examine environment–organism interactions in order to measure plant functions in detail [5]. Another important application of metabolomics, termed as ecological metabolomics, deals with the analysis of plant biochemical connections across distinct temporal and spatial systems. It aims to decipher the possible impact of abiotic/biotic stresses on any vital biochemical process though metabolite identification in response to environmental factors. It explains the biochemical nature of many important ecological phenomena, like the effects of parasite load, disease occurrence, and infection. It also helps to evaluate the interaction among two trophic levels or multiple impacts of abiotic factors with intra- and interspecific linkage. Variations in the concentration of numerous metabolites can give mechanistic indications for biochemical networks that explain the phenotypic and physiological feedback of plants to environmental fluctuations [6]. The full potential of ecological metabolomics is yet to be explored.

Biotic and abiotic stresses adversely affect crop productivity and cause a massive reduction in the global annual crop yield. Metabolomics tools can be integrated with other omics tools, such as genomics, transcriptomics, and proteomics, to tackle abiotic/biotic stresses in plants [7]. It helps to analyze various exogenous and endogenous plant metabolites under extreme climatic stresses and is key to understanding the systems biology of plants. Applications of omics-based strategies to understand the stress regulation process from the genome to phenome are illustrated in Figure 1. The plant metabolome is comprised of two types of metabolites: Primary and secondary metabolites. Metabolic profiling of primary and secondary metabolites provides extensive knowledge of biochemical processes that occur during plant metabolism [8]. Some primary and secondary metabolites of plants can be interlinked with highly complex metabolic pathways. The successful detection, identification, assessment, and evaluation of these metabolites is possible through advanced metabolomics tools, such as gas chromatography-mass spectrometry (GC-MS), liquid chromatography mass-spectroscopy (LC-MS), and non-destructive nuclear magnetic resonance spectroscopy (NMR) [9].

Primary metabolites are important for the biosynthesis of lipids, sugars, and amino acids in plants. They mediate the tricarboxylic acid and glycolysis cycle during photosynthesis and thereby affect plant growth and development. Variations in the synthesis of primary metabolites may lead to photosynthesis malfunctioning and imbalanced osmotic adjustment in plants. Primary metabolism results in the production of secondary metabolites, like flavonoids, atropine, carotenoids, and phytic acid. These are not essential for plant survival and are produced in response to different stress conditions, such as high temperature, chilling, drought, salinity, and insect/pest attack. Additionally, secondary metabolites also include reactive oxygen species (ROS), antioxidants, and co-enzymes [10]. The plant metabolome also consists of some specialized secondary metabolites, such as phenolics (~10000), alkaloids (~21000), and terpenoids (>25000), that offer tolerance against biotic/abiotic stresses. Recently, many of these specialized compounds have been discovered as unique biomarkers that measure plant performance under stress environments and serve as vital components for many crop improvement programs [9]. During plant ontogenesis, primary and secondary metabolites are synthesized continuously via complex biochemical reactions. Therefore, it is essential to uncover the unique metabolic biochemical processes involved in plant biology [11].

Based on the previous research knowledge, numerous methods have been devised for the detection and identification of specific metabolites [12]. However, due to the diverse chemical composition, massive production in cellular compartments, and complex nature of metabolites, no single metabolomics tool can be used for whole metabolome profiling. Although several methods of metabolome extraction and combination of analytical tools are frequently applied for successful metabolic profiling [12], advanced analytical platforms for decoding the whole metabolome of any plant species are still in the non-trivial phase [7]. Metabolomics has promising prospects to expedite the selection of improved breeding materials and screen elite crop varieties. Integration of metabolomics with modern plant genomics tools, such as genotype-based sequencing (GBS), genome-wide genetic variants, and whole genome sequencing, opens exciting horizons for crop improvement [13].

Metabolomics tools are used for metabolic profiling of biofluids and various cell tissues, which are involved in different cellular processes, thus portraying the whole physiological composition of a cell [14]. These can also be used to probe the prototype of any vital gene in plants and permit a genotypic-metabolic-phenotypic level of understanding. As compared to any other living organism, a diverse range of plant metabolites have different orders of instability, size, polarity, solubility, adaptability, and volatility [15,16,17,18]. Metabolomics, metabolic, or metabolite profiling terms are alternatively used to describe three kinds of approaches, such as targeted metabolomics, semi-targeted metabolomics, and untargeted metabolomics [1,7]. Factors like quantification and detection of metabolites, accuracy of the experiment, methods for sample preparation, and evaluation of specific targets are pre-requisites of efficient metabolic profiling. In the case of an untargeted approach, detection of the structural and chemical composition of metabolites is essential while performing targeted and semi-targeted studies enables the chemical nature of metabolites to be evaluated before data procurement [19].

This review describes the advantages and disadvantages of advanced analytical tools and adopted workflow in plant metabolomics research. Here, we also discuss plant genetics and biochemistry implications, which trigger the extraordinary diversity of plant metabolites and provide an overview of the stress tolerance mechanism in plants under extreme climatic conditions. We further focus on the fast-growing era of metabolomics-assisted breeding for crop improvement and the pitfalls in current metabolomics studies are also outlined.

## 2. Advanced Tools for Analytical Research in Plant Metabolomics

The elucidation of plant metabolites in metabolic profiling is considerably challenging due to an insufficient connection between the proteome and metabolome. Additionally, it is hard to detect some metabolites during the whole metabolome analysis due to technical hurdles, like the incompatibility of instruments, lack of standardized protocols, and volatility of the desired metabolites [13]. In metabolomics, no single technique or tool can be used to analyze all the metabolites present in a metabolome; instead, a set of different technologies are required to provide the greatest amount of metabolite coverage. Different metabolomics techniques include mass spectrometry (MS) [20], non-destructive nuclear magnetic resonance spectroscopy (NMR) [21], high-performance thin-layer chromatography (HPTLC), capillary electrophoresis-mass spectrometry (CE-MS) [22], gas chromatography-mass spectrometry (GC-MS) [23], liquid chromatography-mass spectrometry (LC-MS) [24], direct infusion mass spectrometry (DIMS), ultra-performance liquid chromatography (UPLC), high-resolution mass spectrometry (HRMS) [25], and Fourier transform ion cyclotron resonance mass spectrometry (FI-ICR-MS) [26]. Out of these, CE-MS, LC-MS, GC-MS, and NMR-based integrated approaches have been extensively applied for metabolomics analysis. Advantages and disadvantages of some common analytical techniques used in metabolomics are described in Table 1.

The choice of a metabolomics approach depends upon the accuracy, selectivity, speed, precision, and sensitivity of any analytical tool. For a comprehensive study of metabolites, the NMR technique can be used conveniently in many organisms, including plants. NMR-based metabolic profiling is quick, expedient, and efficient technology for the screening and identification of similar biological samples. It is non-destructive, selective, and very proficient at mapping metabolic pathways [27]. Moreover, its high reproducibility makes it a powerful tool in plant metabolomics research. NMR-based metabolic profiling efficiently monitors plant responses under biotic/abiotic stresses at various developmental stages [18]. Many integrated techniques of NMR have been applied to identify the structural units of unknown metabolites. Recently developed advanced tools for plant metabolomics are one- and two-dimensional NMR [28], isotope-labeled NMR [29], and micro-coil NMR [30]. NMR is the only tool that can detect the specific labelling of stable isotopes [3]. It can track molecules with a non-zero magnetic field with one or more atoms, such as 31P, 15N, 14N, 13C, and 1H, and permits metabolite detection by receiving only one signal [31]. NMR is a non-invasive, rapid, highly quantitative, and unbiased approach that requires minor sample preparation and no need for a chromatography separation process. It helps to probe compounds that are associated with insoluble polymers via solid-state high-resolution NMR [32]. As compared to MS, NMR has a lower dynamic range, less resolution and poor sensitivity, resulting in limited coverage of primary and secondary metabolites in plant metabolomics research [33]. However, recent inventions, like the development of miniaturized radiofrequency coils, multi-dimensional NMR techniques, superconducting magnets [34], and cryogenic probes [35], have overcome major limitations in NMR technology.

The mass spectrometry technique provides the benefit of quick sample preparation and examination in their natural state [17]. Ultra-high-performance liquid chromatography (UPLC) and high-performance liquid chromatography (HPLC) are conventional tools for metabolite analysis. However, integration of these tools with mass spectrometry provides efficient analytical platforms for plant metabolome profiling [36]. GC-MS has been recognized as a high-throughput analytical technology with a high rate of sensitivity for metabolic profiling. It offers extremely exceptional detection, separation, and identification due to the application of an electronic impact (EI) ionization point of supply. GC-MS can also be used to probe primary metabolites, like amino acids, organic acids, sugars, alkaloids, lipids, ketones, esters, peptides, and sugar-phosphate. The advantages of GC-MS include its precision, high sensitivity and resolution, reduced running cost, and speedy metabolic profiling [37]. However, GC-MS can only be used to detect thermally unstable and volatile compounds [38].

In contrast to the GC-MS approach, LC-MS uses an electrospray ionization (ESI) source to analyze high molecular weight metabolites, which are polar and thermo-labile. It is executed largely for secondary metabolite profiling, including vitamins, glucosinolates, flavonoids, and carotenoids, but can also be used for primary metabolites’ detection. LC-MS has unique features that permit direct probing of metabolites in any sample with no need for derivatization [39]. Moreover, LC-MS-based metabolic profiling is performed by both targeted and non-targeted methods. In the targeted technique, a set of metabolites are identified and quantified while in the non-targeted approach, numerous types of chemical compounds, like lipids, amino acids, and their derivatives, are detected [1]. In plant metabolomics, integrated approaches of LC and MS are extensively used for analytical research due to their greater accuracy and sensitivity [12]. Advanced tools of mass spectrometry, such as desorption electrospray ionization mass spectrometry (DESI) and matrix-assisted laser desorption ionization (MALDI), have been applied to achieve high-resolution imaging in metabolomics that indicates the arrangement patterns of metabolites in plant cells and tissues [40]. With recent integrated applications of MS approaches, metabolomics has emerged as a more versatile strategy than genomics and proteomics. Indeed, metabolic profiling of various crops, such as wheat, rice, maize, sorghum, and soybean, showed remarkable applications of metabolomics in plant biology [4,41,42,43,44].

## 3. The Workflow of Metabolomics Analysis

The core steps involved in the metabolomics experimental design are sample preparation, data acquisition through analytical strategies, and utilization of suitable chemo-metric techniques for data mining [45]. These steps are described briefly:

### 3.1. Sample Preparation

Sample preparation is one of the most crucial parts of metabolomics as it has a tremendous impact on the final results [45]. Aboveground plant tissues, such as seeds, stems, and roots, can be used as sample material [46]. In plant metabolomics experiments, thee high-resolution magic-angle spinning technique is widely used even though it is not suitable for the extraction of plant secondary metabolites that play an important role in plants’ self-defense mechanism [47].

The main objective of sample preparation is to separate metabolites from unwanted elements and enrich the desired metabolites. The best sample preparation technique should be quick, economical, simple, easy, and uphold the sample integrity [48]. Plant sample preparation for metabolic analysis involves four steps: Harvesting the plant material, quenching, extraction, and sample analysis. Depending upon the characteristics of the metabolites and the choice of analytical methods, the extraction and freezing steps can be omitted. Additionally, the harvesting and quenching of the sample material are the same for all analytical tools but quenching of the sample material is subjected to the biological nature of the sample. Harvesting of the sample must be performed cautiously, as the plant metabolome is sensitive to enzymatic reactions, which degrade various metabolites. Usually, the plant material is quenched in liquid nitrogen immediately after harvesting to avoid any metabolic changes [45]. Similarly, the age of the plant sample is also very important as metabolic profiling of young leaves is quite different from mature leaves. Avoiding enzymatic degradation of the sample material is very critical for sample preparation [49].

Many extraction protocols have been developed in the last few years for metabolomics analysis [46,50]. Conventionally, a pestle and mortar are used for grinding leaves [51], and these have now been replaced with other methods, including an electric grinder, tissue lyser, and ultrasonic oscillator. The choice of extraction solvent is also very important in metabolite extraction. The solvent should have no signals and be easily separated without triggering any biochemical reaction. Aqueous methanol, ethanol, perchloric acid, acetonitrile, and water are commonly used as extraction solvents [46]. The choice of extraction protocol depends upon its rate of dissolution and solubility. Biological components, such as cellulose or lignin, might interact with metabolites and consequently affect the dissolution rate. One of the conventional methods employed for sample extraction is Soxhlet extraction. In this technique, the sample is heated continuously, and concentrated solvent is used for extraction. Solid phase microextraction is an important method used extensively for targeted and untargeted metabolic profiling via mass spectrometry approaches [52]. Laser microdissection (LMD) is an excellent technique to isolate the desired cells from microscopic samples. It does not affect the chemistry and morphology of the desired metabolites in the samples [53]. Microwave-assisted extraction (MAE) is another high-speed and accurate method of sample extraction in metabolomics [46]. For volatile metabolites, an efficient method called supercritical fluid extraction can be used [54]. Many other sample preparation methods exist, including ultrasound-assisted extraction (UAE) [55], enzyme-assisted extraction [56], solid phase microextraction (SPME), and the Swiss rolling technique [57]. With new innovations in metabolomics, novel methods for extraction are emerging day by day in accordance with the selection of analytical techniques and nature of the metabolites.

### 3.2. Data Mining, Annotation, and Processing in Metabolomics

The modern era of metabolomics provides deep insights into the molecular complexity of downstream of the genome, transcriptome, and proteome of plants either in normal growth conditions or in response to various stresses. Whole metabolome analysis has established an enormous amount of data sets because of the huge number of metabolites present in different parts of plant cells or tissues. The complicated nature and composition of metabolites in different plant samples has made metabolomics data analysis more complex. The basic goal for whole metabolome analysis is to categorize the different metabolites of various plant samples induced by numerous factors [58]. A considerable amount of data can be generated by metabolomics, so it is called the data-rich technique [59]. Effective metabolomics analysis relies on both the wet and dry science [60]. Powerful automated tools are necessary to manage huge datasets, and annotate and store the raw data [61]. The real challenge in plant metabolomics is to extract accurate information about specific metabolites from massive datasets generated by advanced techniques. Basic steps involved in data mining include pre-processing, pre-treatment, and statistical analysis of data [62,63]. Therefore, sophisticated statistical tools are needed for rapidly pinpointing and measuring all targets in a sample. Raw data acquired from sample analysis is subjected to a series of statistical analyses to generate a numerical data matrix and align this data for further processing.

### 3.3. Statistical Tools and Biomarker Identification

Metabolomics measures the metabolite abundance as a predictive biomarker for disease diagnosis. It also scores the genetic as well as environmental-induced changes in metabolites’ concentration. The identification of biomarkers relies on data analysis using different statistical methods. Metabolic marker probing is connected to the concept of linking response variables, such as the desired phenotype, to explanatory variables representing biomarkers. An appropriate, multi-dimensional statistical platform is mandatory for fast forward analysis in order to estimate the relationship among metabolites and phenotypic variables. A pairwise Pearson’s correlation can be used to discover a specific biomarker, where only one metabolite is connected to the desired phenotype. Although, more than one metabolite analysis is required to design a predictive model, and canonical correlation analysis (CCA) is often applied to study the maximize correlation between variables [64,65].

Statistical tools can be used to handle the high-throughput metabolomics data that are mainly adapted from already existing omics technologies. Many statistical tools originally developed for transcriptomic analysis can also be used for metabolomics data analysis. Traditionally, in any metabolomics data, we aim to see the groupwise differences either in a univariate, i.e., parameter-by-parameter fashion using univariate techniques (t-test, analysis of variance (ANOVA), and Mann–Whitney U-test). Univariate analysis is generally performed for biomarker discovery at initial levels of systems biology, which studies one variable at a specific time. In addition, it can also verify the performance and legitimacy of a presumed metabolic marker [66]. On the other hand, multivariate analysis can be employed for screening plant cultivars and ecotypes, disease diagnosis, and metabolic marker discovery [2]. These tools have been used for efficient comparative evaluation among different genotypes and samples. Multiple data fusion strategies help to explore the variations at the molecular to phenotypic level of plants [67]. There are many multivariate statistical tools available, including ANOVA, analysis of variance-simultaneous component analysis (A-SCA), principal component analysis (PCA), partial least squares-discriminant analysis (PLS-DA), and heat map analysis. The selection of statistical tools should be made according to the experimental scheme of the procedure [68].

In order to study high-throughput metabolomics data, multivariate statistical strategies are generally categorized into two approaches [69]: Unsupervised approach in which unidentified samples are statistically analyzed, focusing on the natural structure existing in a data set; and supervised techniques, which aim to alter multivariate datasets from metabolic analysis to the demonstrations of biological units under supervision, sometimes known as predictive platforms [70]. The supervised method describes the relationship among the input and output observable in a specific sample of data.

PCA is recognized as one of the most important unsupervised multivariate statistical tools and is being extensively used for the multi-dimensional reduction approach [71]. Furthermore, PCA is beneficial and efficient because the variation among different treatments or samples can be divided and comprehensively explained in numerous principle components. Although, PCA is unable to separate variance in samples whenever a multi-purpose factor is strongly connected or co-occurrs [71]. On the other hand, the PLS approach has the advantage of managing noisy and highly collinear data sets. Moreover, extensions of PLS, like orthogonal PLS (OPLS), sparse PLS (sPLS), and PLS-DA, are also frequently performed in metabolic data analysis. OPLS and PLS methods give significant information that can be useful for metabolic marker selection [72,73]. Commercially available statistical tools are Matlab (https://www.mathworks.com) and SIMCA-P (http://umetrics.com/products/simca), which propose several types of procedures. Several other packages established for metabolic profiling [74,75,76,77] are shown in Table 2. As these statistical tools are applied in numerous fields, they can be used in several statistical software applications, which are not specially programmed for metabolic profiling. Some excellent R programming software has been developed for various applications in plant metabolomics research. The R package language statistical tools are designed to give statistical graphics and computing, and a large number of statistical analysis techniques are employed in R package programs [78]. For univariate and multivariate metabolomics analysis, R package can be used to guide the user in the most convenient way through a step-by-step pipeline from pre-data processing to data assessment, evaluation, interpretation, and visualization, to the detection of interesting metabolites. Recently, some excellent R software packages have been designed for reproducible and flexible data analysis, pathway-based modelling, and liner modelling for quantitative data analysis. MetabR (http://metabr.r-forge.r-project.org/) [79], MetaboAnalystR (https://github.com/xialab/MetaboAnalystR) [80], Lilikoi (https://github.com/lanagarmire/lilikoi) [81], and MetaboDiff (http://github.com/andreasmock/MetaboDiff/a) [82] are some important R software packages available for metabolomics analysis.

### 3.4. Bioinformatics Tools and Databases

Time is no longer a limiting factor for data mining in metabolomics due to the rapid advancements in modern analytical and technical tools. Computational informatics is a prerequisite of metabolomic experiments [83]. During the past few years, numerous online web-based programs have been designed to aid metabolomics data mining, data assessment, data processing, and data interpretation. The disposal of accurate and economic assessable platforms has enormously eased the design and maintenance of web tools that can be used by many researchers with little bioinformatics skills and limited computational facilities [84]. However, handling the huge raw datasets frequently via the internet poses substantial drawbacks. XCMS is an online bioinformatics tool (https://xcmsonline.scripps.edu), which allows raw data to be uploaded directly and assist the user in statistical analysis and data processing [85]. XCMS servers are unable to deal with the massive data files due to limited space. Recently, the XCMS stream has been established for programmed data transfer in LC-MS experiments that reduced the data processing time and enhanced the efficacy of an online system [86]. It can also be used to detect mutative substances via MS tools applying the METLIN database [87]. In addition, R scripts are programmed in-house to get output comprising of characteristics in the formation via the XCMS package, which can be applied to execute statistical investigation and metabolite detection through the MS/MS database and formula predictor [88].

METLIN (http://metlin.scripps.edu) is another online database applied in numerous studies related to stress response metabolic profiling in plants. It is beneficial for metabolic profiling of unique metabolites and the workflow of data mining, annotation, and processing is not time-consuming. Furthermore, it permits immediate retrieval of LC/MS, MS/MS, and FTMS analysis outcomes by allowing its operator to put a query in the database through a programmed framework [89]. MetaGeneAlyse (http://metagenealyse.mpimp-golm.mpg.de/) is a web-based tool that implements regular clustering techniques, like independent component analysis (ICA) and k-means. This web-tool also offers many ways for statistical analysis, such as pathway enrichment analysis, PLS-DA, and t-test [90]. During the last decade, MetaboAnalyst has been modified efficiently by integrating various tools, like MetPA (http://metpa.metabolomics.ca) [91] and MSEA (http://www.msea.ca) [92]. A significant web-based platform that has been employed in plant metabolomics for data assessment, processing, and statistical analysis is MeltDB (https://meltdb.cebitec.uni-bielefeld.de) [93]. Other databases, such as iMet-Q (http://ms.iis.sinica.edu.tw/comics/Software_iMet-Q.html), MS-Dial (http://prime.psc.riken.jp/Metabolomics_Software/MS-DIAL/), and MetAlign (www.metalign.nl), are operated by windows GUIs (graphical user interfaces) and do not need any local installation [94,95,96]. To study the metabolome with a species-nonspecific or species-specific origin, MZedDB and KEGG (http://www.genome.jp/kegg/) have been extensively applied [97,98]. Recently, a new tool, Galaxy-M, has been developed to examine untargeted metabolites through LC-MS techniques [99]. Meta box is another online server and has numerous applications in the elucidation of omics data [100]. Babelomics (http://www.babelomics.org/) [101] and GenePattern (http://software.broadinstitute.org/cancer/software/genepattern/) [102] are two omics-based web tools that have been used to perform univariate and multivariate statistical analysis, gene expression data interpretation, and visualization of metabolomics data. Table 2 shows the various statistical, web-based, and online bioinformatics tools extensively used in plant metabolomics for data analysis [103,104,105,106,107,108,109,110,111,112,113,114,115,116,117,118,119,120,121,122,123].

## 4. Metabolomics for Crop Improvement

Metabolomics has emerged as the most promising tool to decipher abiotic stress tolerance in plant species. Recently, metabolomics has been applied to probe for unique metabolites during the life cycle of plants. Biotic/abiotic stresses have a significant role in the reduction of the crop yield [16]. Plants respond to both stresses in a similar mechanism, but these stresses produce variations in plants’ biochemical and physiological processes. The detection of invading organisms depends upon the recognition of specific molecular units. The invaders trigger immune sensors in plants, providing resistance, including effector-triggered immunity (ETI), pattern-triggered immunity (PTI), and termed pattern recognition receptors (PRRs) [124]. At the onset of abiotic stress, the plant synthesizes phytohormones to impart stress resistance [125]. The oxidative stress disturbs the stomatal conductance and activates several signaling mechanisms [126]. Overall, the specific gene expression profile depicts the exact composition of metabolites in a particular plant species. A novel bioactive agent is synthesized due to the activation of a certain metabolic network [125]. A flowchart demonstrating the general steps involved from diagnostics to metabolomics-assisted breeding for crop improvement is shown in Figure 2. In the following section, we elaborated the role of metabolomics in deciphering biotic and abiotic stress tolerance in crop plants.

### 4.1. Elucidation of Abiotic Stress Tolerance in Plants

Abiotic stresses are major limiting factors of agriculture production and can be described as any change in plants’ growth conditions that adversely affect plant metabolism, normal plant development, and plant physiology. The main abiotic stresses that negatively regulate the plants’ growth are drought, salinity, temperature extremes, waterlogging, heavy metal, and chilling [127]. Metabolomics has emerged as the most promising tool for investigating abiotic stress tolerance regulation in plant species. Recently, metabolomics has been applied for probing unique metabolites that regulate the abiotic stress tolerance mechanism in crops. The prime objective of investigating metabolic variations under abiotic stresses is to detect different metabolites that permit restoration of plant homeostasis and normalize metabolic modifications. Furthermore, it is also used to probe specific compounds responsible for offering abiotic stress tolerance in plants [11,14].

Many advanced tools, like NMR, LC-MS, and GC-MS, are extensively employed in metabolomics studies to elucidate abiotic stress tolerance in plants [18]. GC-MC can generally be used for plant metabolic profiling under abiotic stresses. Time of flight (TOF-MS) also provides fast and efficient differentiation and detection of various metabolites in mixed samples or treatments [11,17,128]. For the identification of abiotic stress-regulated metabolites, Fourier-transform ion cyclotron resonance, linear trap quadrupole (LTQ), ion trap, triple quadrupole, and quadrupole TOF (qTOF) have been employed [129]. All essential mechanisms in plants from germination to maturity are severely affected by abiotic stresses [17]. Under drought and salinity stress, plants face osmotic stress by disordering the ion concentration and homeostasis [130]. Abiotic stresses badly affect plant photosynthesis as well as hamper the synthesis of all primary metabolites, including amino acids, sugar alcohols, and sugars [131]. Recent advancements in plant metabolomics to identify abiotic stress tolerance in major crops are summarized in Table 3.

#### 4.1.1. Drought Stress Regulation

Drought is a major limiting factor for agricultural production around the world. In drought stress, plants adopt several physiological modifications, such as leaf abscission, lead area reduction, and greater nutrient uptake by plant roots. Furthermore, stomatal closure affects transpiration activity and reduces water loss. These physiological alterations improve water use efficiency (WUE) that ultimately hinders photosynthesis activity due to reduced CO_2_ concentrations and stomatal closure [132]. Plants synthesize many ubiquitous polyamines, like spermine, spermidine, and putrescine, in response to drought stress as a defense mechanism [133]. Metabolomic profiling was conducted for six wheat genotypes under drought stress. These drought-tolerant genotypes produced several important metabolites, such as γ-aminobutyric acid (GABA), myo-inositol, threonine, proline, oxalic acid, malic acid, glucose, fructose, and sucrose [134]. Comparative metabolic analysis of drought-tolerant and -susceptible wheat cultivars indicated the accumulation of high levels of lysine, arginine, methionine, and proline in response to drought stress [135]. Similarly, accumulation of linamarin, pipecolate, proline, and tryptophan has also been detected by metabolomics analysis of wheat under drought stress and these can be used as potential biomarkers to screen drought-tolerant genotypes [136].

Recently, GC-MS-based metabolic profiling of eight wheat cultivars was performed to observe the drought tolerance mechanism. Data regarding the chlorophyll content, stomatal conductance, relative water content, and metabolites of different cultivars was recorded. An increased concentration of glutamine, serine, methionine, lysine, and asparagine was recorded under drought stress [20]. In 2018, Yang and colleagues carried out metabolic profiling of maize under drought stress via RP/UPLC-MS, indicating upregulation of lipid and carbohydrate metabolism and acceleration of the glutathione cycle [14]. Furthermore, LC-MS- and GC-MS-based metabolic profiling confirmed the differential accumulation of metabolites in young and matured leaves [137,138]. In another study, GC-MC-based metabolic fingerprinting of 10 maize hybrids was performed under drought stress. The results revealed low concentrations of sugars, such as erythritol and maltose while no change was observed in the concentration of xylitol and raffinose during drought stress. Drought-tolerant maize cultivars hyperaccumulated GABA, leucine, glycine, serine, alanine, and tryptophan [4]. Enhanced production of 4-hydroxycinnamic acid, ferulic acid, stearic acid and xylitol was recorded using a GC-MS tool in rice under drought conditions [139]. A combined strategy of GC-TOF-MS-based metabolic analysis has been carried out in rice to study the effect of drought stress in polyamine metabolism. The results showed well-coordinated regulation of polyamine biosynthesis, thereby increasing the synthesis of spermine to regulate drought tolerance in rice [140]. Metabolomics profiling of different crops, like wheat, barley, rice, and soybean, has been carried out to elucidate vital metabolites offering drought tolerance [16,17,141,142,143].

#### 4.1.2. Salinity Stress Regulation

A higher level of soil salinity results in ion toxicity and disturbance of the ion uptake mechanism, causing metabolic syndrome and osmotic imbalance that leads to stunted growth and the capture of several physiological activities [144]. Imbalanced Na+ ion concentrations cause ion toxicity, which hampers nutrient and water uptake in high salinity conditions [145]. Plants synthesize many primary and secondary metabolites to cope with salinity stress conditions.

The GC-MS-based metabolic profiling of rice seedlings was performed under salt stress that proved hyperaccumulation of important amino acids, like leucine, isoleucine, valine, and proline [15]. Gupta and colleagues carried out comparative metabolic profiling of tolerant and sensitive rice by using the GC-TOF-MS technique. Elevated levels of amino acid accumulation in tolerant genotypes were observed as compared to susceptible varieties. [146]. GC/MS-based metabolic profiling of barley roots under salt stress conditions revealed an increased production of organic acid, proline, sucrose, xylose, and maltose, which showed the salt tolerance mechanism in barley [147]. In a recent report, metabolic analysis of rice under salt stress indicated enhanced concentration of sucrose and mannitol, and lower contents of quinate and shikimate, which provide salinity tolerance in rice [23]. Modifications in the abscisic acid (ABA) pathway can disturb other hormonal pathways associated with salt stress. For example, the jasmonate (JA) pathway is a significantly important hormonal pathway that is receiving attention presently. It was reported that the application of jasmonates reduces salinity damage in rice species [148]. Moreover, various metabolomics tools have been employed to study the variations in the metabolic profile of many crop plants, like tomato, maize, barley, and wheat [129,149,150].

#### 4.1.3. Waterlogging Stress Regulation

Waterlogging is another type of abiotic stress that hinders crop growth and yield. Waterlogging causes extreme injuries to plants due to the limited supply of CO_2_ and oxygen, which ultimately hampers the photosynthesis process. Waterlogging for a longer time causes hypoxia that directly affects roots and prevents CO_2_ assimilation. Signal transduction, metabolic alteration, and morphological changes are three adaptations to waterlogging stress [151]. NMR-based metabolomic profiling of soybean indicated a negative correlation of waterlogging stress to the synthesis of primary and secondary metabolites [152]. In another experiment, metabolic profiling of soybean root tips under waterlogging was performed with capillary electrophoresis MS that unveiled 73 metabolites. The concentrations of phosphoenolpyruvate, nicotinamide adenine dinucleotide (NADH2), glycine, and gamma-aminobutyric acid were elevated under waterlogging conditions [22]. Herzog and colleagues used GC/MS and LC/MS to study the submergence tolerance mechanism in two wheat cultivars, identifying lysine, proline, methionine, and tryptophan as important biomarkers for waterlogging tolerance [153]. In another experiment, the resilience of different rice cultivars under submergence stress was investigated via metabolic profiling. In this study, NMR- and GC/MS-based integrated metabolomics tools detected some unique metabolites, including 6-phosphogluconate, phenylalanine, and lactate, that allowed rice plants to tolerate waterlogging stress conditions [154].

#### 4.1.4. Temperature Stress Regulation

Plants need optimum temperatures for their normal growth and development. Fluctuations in temperature can cause severe damage and cease the developmental processes. High temperatures disturb the homeostasis and other physiological mechanisms [25,155]. Plants synthesize many secondary metabolites under heat stress, such as rhamnose, putrescine, myoinositol, 2-ketoisocaproic acid, arachidic acid, allantoin, and alanine [155]. Untargeted metabolomics of wheat was performed by using LC-HRMS under heat stress. Metabolic profiling showed a significant rise in metabolite synthesis, such as pipecolate and L-tryptophan, under heat stress. Moreover, heat stress affects the biosynthesis of secondary metabolites and biosynthesis of aminoacyl-tRNA [25]. The metabolic profiling of what grains using LC-MS/MS-HPLC technologies indicated higher concentrations of sucrose and G1p under heat stress [156]. Comparative metabolomics analysis was carried out for heat-tolerant and -susceptible soybean cultivars. The heat-tolerant genotypes synthesized higher concentrations of carbohydrates, including 1,3-dihydroxyacetone, ribose, and glycolate, as compared to the susceptible ones. These tolerant types also produced low concentrations of many metabolites, such as chiro-inositol, pinitol, erythritol, and arabitol [157]. Metabolomics analysis was also performed for other important crops, such as tomato, maize, and wheat, to observe the effects of heat stress [158,159,160].

#### 4.1.5. Metal-Induced Stress Regulation

Heavy metal stress is another abiotic stress that has become a very important factor that influences crop yield. It prompts several signals in plants, heading towards variations in complex biochemical, molecular, and physiological mechanisms. Trace elements, such as cobalt Co W, V, Cr, Zn, Ni, and Cu, are considered lethal to plants at their higher concentrations [43]. Metals, like chromium (Cr), nickel (Ni), zinc (Zn), lead (Pb), cesium (Cs), and cadmium (Cd), are regarded as major pollutants influencing plant stress. Metal stress imposes metabolic retardation, cellular oxidation, and enzyme inhibition that results in growth arrest and even cell death in higher concentrations [161]. Similarly, copper (Cu), manganese (Mn), and iron (Fe) are vital for plant growth in many ways due to their importance in numerous biological processes. Plant reproduction, metabolism, and growth have been adversely affected by heavy metals. To combat the heavy metal stress, plants require multi-dimensional biochemical and physiological synchronization, stabilization in protein structure, and modifications in the whole metabolome in order to develop metal stress tolerance. Metabolic analysis of beans showed elevated concentrations of carbohydrates under higher treatments of Zn and Cu [162]. Heavy metal stress led to hyperaccumulation of metabolites in *Brassica rapa*, as indicated by 1H-NMR, 2D-J-based metabolomic analysis [163]. Similarly, CapHPLC-ESI-QTOF-MS-mediated metabolomics profiling of sunflower roots and leaves indicated the enhanced production of special kinds of fatty acids under Cr metal stress [161].

#### 4.1.6. Nutritional Deficiency Regulation

For normal growth and development, plants require essential nutrients in optimum concentrations. Organic molecules and metabolites are built of structural units, like carbon, phosphorus, sulfur, and nitrogen, in plant cells. Deficiencies of these nutrients directly affect the growth, metabolism, and physiology of plants. Nitrogen (N) is considered as the most vital element in nature as it is the basic structural unit for cellular metabolites, such as nucleic acid, amino acids, and proteins, as well as for many secondary metabolites [164]. Plants synthesize metabolites, such as threonate, glycerate, inositol, and several soluble carbohydrates, under a limited supply of nitrogen [165]. Khan and colleagues carried out wheat metabolomic profiling based on GC-MS and LC/MS technology. They reported a higher production of tyrosine, lysine, allo-inositol, and L-ascorbic acid in wheat under N stress [11]. Similarly, higher concentrations of fructose, ribulose, and lyxose were observed in wheat metabolome analysis via an integrated technique of GC-TOF-MS under N stress. [166]. Metabolic fingerprinting has been carried out in tomato (*Solanum lycopersicum* L.) under N and phosphorus (P) deficiencies. It was observed that N stress reduces the concentration of organic acids and amino acids while triggering the synthesis of soluble sugars [8]. The second most important macronutrient, sulfur (S), is a pre-requisite for the synthesis of many important metabolites and amino acids containing sulfur as a structural unit [167]. Recently, Ghosson and colleagues applied a UHPLC-mediated metabolomics platform to study the effects of S stress on the roots and leaves of barley. Different amino acids, organic acids, and S-responsive metabolites were synthesized due to nutrient stress [168]. P is essential for the synthesis of DNA, RNA, ATP, and numerous metabolites involved in energy metabolism [169]. Metabolic profiling of barley exposed to P deficiency indicated that plants produced di- and trisaccharides while the concentrations of many organic acids and P-containing intermediates was reduced [170]. Similarly, metabolic fingerprinting of the common bean was performed to study the effect of P stress in nodules and roots [171] and low N in wheat [172].

### 4.2. Elucidation of Biotic Stress Resistance in Plants

Metabolomics profiling determines important changes in plant primary and secondary metabolites due to any pathogen attack [173]. Plants adopt numerous strategies to trigger defensive pathways against pathogen attack. The presence of highly diversified metabolites in plant cells makes it difficult to decode the whole metabolome of a plant species [174]. Plants have accumulated several metabolites in response to biotic stresses that are tissue and species specific and act as biomarkers to regulate biotic stress resistance in various plant species [175]. Metabolomic profiling indicated benzoxazinoids (BXs) accumulation in grasses. BXs are important secondary metabolites serving as a defense mechanism against biotic stress. In maize, BXs have been detected in large concentrations [176]. Moreover, plant–microbe association can also produce resistance against pathogen attack by inducing several associated molecular mechanisms. This includes the biosynthesis of complex plant metabolites and their systematic signaling in the cell. These metabolites have been isolated from the model plant, *Arabidopsis thaliana*, and investigated for their resistant ability in barley. The result showed an improved disease resistance in barley as compared to wild cultivars [177]. Comparative metabolic profiling of infected and healthy plants identifies complex metabolic networks related to plant–pathogen interaction [178]. Phenylpropanoids are a vital component of lignin, which is the basic unit of the cell wall, and plants respond to pathogen attack by modifying the cell wall composition and root architecture. A thick cell wall may assist in protecting plants against pathogen attack. Increased synthesis of phenolic acid and phenylpropanoids were observed in wheat under the attack of *Fusarium graminearum* [179].

Recently, Seybold and colleagues analyzed the wheat metabolome to elucidate the stress responsive mechanism against *Zymoseptoria tritici*. Microbiome and comparative metabolomic profiling were conducted via FT-ICR-MS to detect immune and defense-related metabolites in resistant and susceptible wheat cultivars [26]. The wheat metabolome was studied to discover the unique biomarkers against *Fusarium graminearum*. NMR-based metabolic analysis indicated the higher accumulation of some disease-resistant biomarkers, such as trehalose, asparagine, phenylalanine, myoinositol, 3-hydroxybutarate, and L-alanine [21]. Likewise, in an effort to identify other disease-resistant biomarkers in wheat against *Fusarium graminearum*, metabolic analysis was conducted via NMR and revealed the enhanced production of spermine, putrescine, GABA, inositols, galactose, and lactic acid [180]. Comparative metabolomics analysis has been carried out to study the metabolome of resistant and susceptible varieties of wheat infected by wheat streak mosaic virus. UPLC-QTOF/MS was used for metabolic profiling and indicated the reduction of some metabolite signals, which ultimately deactivated the wheat streak mosaic virus signals [41].

In 2016, Suharti and colleagues performed metabolomics analysis via the CE/TOF-MS platform to examine the resistance response of rice cultivars against *Rhizoctonia solani* attack. Jasmonic acid, mucic acid, and glyceric acid were synthesized in greater concentrations in response to fungal infection [42]. GC-MS tool-based metabolic profiling of three rice cultivars attacked by gall midge indicated increased levels of phenylalanine and glutamine with a high accumulation of linoleic acid in resistant varieties [181]. Another study reported hyperaccumulation of lipids, carbohydrates, alkaloids, xanthophylls, and acetophenone in *Xanthomonas oryzae* pv. oryzae (Xoo)-resistant rice cultivars [182]. LC-MS, GC-MS, and NMR-based metabolomics tools have been employed to examine the metabolic profile of rice infected by *Magnaporthe grisea*, which is one of the most damaging pests of rice. The metabolic analysis demonstrated that significant variation has been induced by *M. grisea* in the rice metabolic profile [183]. The maize metabolomics analysis was conducted to investigate the mechanism underlying resistance against *Fusarium graminearum* and identified two metabolites, such as smiglaside and smilaside, that might be linked with resistance against fungal attack [24]. In another experiment, FT-IR and NMR techniques were used to study the disease-resistant mechanism subjected to southern corn leaf blight attack. Metabolomic profiling showed the presence of lignin, flavonoids, and polyphenols that provide resistance against pathogens [184].

*Ostrinia furnacalis* is one of the most damaging insects of maize and causes massive yield reduction of commercial maize on a large scale. Metabolomics analysis revealed the higher production of volatiles, phtohormones, and benzoxzinoids, and upregulation of many metabolic pathways related to disease resistance in maize [185]. The interaction among herbivores’ insects and plants are multi-dimensional and complex. *Nilaparvata lugens* is a destructive insect pest of rice and offers an ideal system to study plant–insect interactions. Metabolic profiling via GC/MS was performed to study the metabolic fluxes in response to insect attack and showed an enhanced synthesis of GABA and glyoxylate, which offers resistance against brown planthopper [186]. In another study, rice metabolomics analysis was executed to study the resistance against *Chilo suppressalis* and aphid resistance in wheat [187,188]. Weeds are also considered as destructive pests for crop yield reduction. Shoot and root extracts were examined to study the allelopathic nature of canola metabolites that inhibit the root and shoot development of ryegrass. Metabolomics analysis detected some useful allelopathic compounds, such as 3,5,6,7,8-pentahydroxy flavones, p-hydroxybenzoic acid, and sinapyl alcohol, that play a crucial role against ryegrass [189]. Targeted metabolomics analysis via LC-MS/MS Q Trap was conducted to probe unique metabolites for weed suppression in different wheat genotypes [190,191]. Similarly, metabolic profiling was carried out to decipher the weed suppressive metabolites in different annual legumes. The UHPLC QTOF-MS-based technique was executed and found an abundance of flavonoids that can be used to suppress different weeds [192].

### 4.3. Soil Metabolomics

Soil metabolomics is another important application of metabolomics for crop improvement. Soil consists of organic/inorganic compounds as well as a wide range of microbial populations, the connection of which is significantly elaborated as soil metabolomics. The relationship among crop production and the soil metabolome is an area of great interest because soil microbe diversity and composition affect crop production. A precise metabolic analysis of soil microbe populations in the soil metabolome and their connection to diverse metabolite synthesis is crucial to increase our understanding and allow the exploration of new avenues in soil microbiome investigations [193]. Metabolic profiling of soil samples from wheat rhizospheres was carried out and numerous bioactive metabolites, such as glutarimide, consabatine, methylpyrrole, arachidonic acid, gibberellic acid, and diacetyllycopsamine, were detected. The results showed that these unique biotic metabolites are involved in many soil–plant signaling pathways and provide a defense mechanism against pathogens [194]. Soil metabolomics analysis was performed to identify suppressive soils for cereal production. LC/MS and 1H NMR tools were applied for soil metabolic profiling of disease-suppressive soils. A unique biomarker, macrocarpal was detected for *Rhizoctonia solani*-suppressive soil [195]. Similarly, NMR-based soil metabolic profiling was done to discriminate the suppressive and non-suppressive soils for disease resistance [196].

## 5. Metabolomics-Assisted Breeding

During the last decade, metabolomics has perceived remarkable developments in both software tools’ design and instrumentation advancement, providing an excellent opportunity to scan the whole metabolome of various plant species in a high-throughput way. The applications of metabolomics have assisted numerous research areas, especially biotechnology, like precision plant breeding, functional genomics, and disease diagnostics [197]. Furthermore, its application makes a way forward for translational metabolomics in plant breeding. Recent advances in post-genomic approaches have accelerated the screening process, and the integration of metabolomics with other high-throughput tools will reduce the time needed to develop elite crop varieties with improved tolerance against abiotic and biotic stresses. Metabolomics has great potential to provide a holistic examination of numerous metabolites’ diagnosis and phenotyping of plants [198]. About 840 metabolite units have been detected in 524 rice cultivars, which have potential for exploitation in future crop breeding strategies [199]. As holistic datasets of transcriptomics, proteomics, and metabolomics are available, scientists are applying these techniques in epigenomic QTL (eQTL), proteomic QTL (pQTL), and metabolic QTL (mQTL), respectively, for quantitative traits mapping and dissecting genetic variations at the mRNA, protein, and metabolic levels, as shown in Figure 3. Genome-wide association studies (GWASs) assisted by metabolomics techniques (mGWAS) and metabolic quantitative trait loci (mQTLs) are powerful tools to detect genetic variations linked with metabolic traits in plants [200].

### 5.1. Metabolic QTLs (mQTLs)

Knowledge about metabolic networks controlling the complex mechanisms in metabolomics have a potential role in metabolomics-assisted breeding to develop elite cultivars for better quality and yield. Furthermore, data obtained from mQTL investigations lead to more comprehensive knowledge about quantitative genetics [201]. Metabolic profiling narrows down the gap between the genotype and phenotype and opens up new horizons for metabolic dissecting, beginning with the identification of single nucleotide polymorphism (SNP) markers or mQTL mapping analysis for candidate gene detection. Metabolic markers are an efficient tool for agronomic trait discovery and exploration of the biological pathways responsible for various phenotypes [65]. The mQTLs approach establishes a linkage between the phenotype and genotype, highlights important insights of the genetic structure, and dissects phenotypic variations through integrated analysis of gene expression and metabolic profiles [202].

Advancement in next-generation sequencing (NGS) has allowed mQTLs identifications for candidate genes through ultra-high-density maps [203]. Candidate genes controlling the biosynthesis of secondary metabolites can be detected by employing multi-omics tools integrated with reverse and forward genetics approaches [204]. Furthermore, population genetics combining quantitative genetics with metabolic profiling has started to uncover genetic control of the whole metabolome in plants. For example, rice mQTL analysis has been performed using a high-density map comprising 1619 bins produced by sequencing. Across 12 chromosomes, many mQTLs have been detected in flag leaf and germinating seeds [200]. Comparative investigations of two rice cultivars for mQTLs analysis revealed the accumulation of tissue-specific secondary metabolites that are under strict genetic control. A total of 19 metabolites were detected on 23 loci, proposing a significant intersection of genetic control in various cells [200]. Likewise, similar results have been reported for tomato [205] maize [201,202] and potato [206]. The mQTL analysis of back-crossed inbred lines (BILs) of rice cultivars observed 700 various metabolic features. The study unveiled 802 mQTLs having an irregular distribution, which might control different metabolic traits [207].

Recombinant inbred lines (RILs) of barley were subjected to mQTL analysis and identified 98 different metabolites under drought stress. These stress-responsive metabolites, including sinapic acid, ferulic, and flavones, serve as antioxidants and exert control over gene expression regulation and modulate protein function under stress [208]. Templer and colleagues investigated barley mQTLs, physiological, morphological, and metabolic adaptation under drought stress conditions. About 57 metabolites and some unique mQTLs, such as succinate, glutathione, and γ-tocopherol, were identified in flag leaf via association genetics. The results showed a molecular basis for barley breeding with increased tolerance against drought stress [209]. In *Brassica napus*, metabolomic profiling and genetic analysis were conducted based on glucosinolate synthesis. Results showed 105 mQTLs linked with glucosinolate production in seeds and leaves [210].

The dissection of genomic regions linked with the synthesis of secondary metabolites in wild and introgression lines (ILs) of tomato led to the identification of 679 secondary mQTLs, which are linked with environmental stress tolerance [211]. In the later studies, tomato mQTL analysis was performed to study different concentrations of metabolites of a similar ILs population [212]. mQTL mapping is an efficient tool for the identification of traits associated with stress susceptibility. The LC/MS-mediated metabolomics profiling of 179 doubled haploid wheat lines probed about 558 secondary metabolites, including phenylpropanoids, flavonoids, and alkaloids [213]. GC-TOF/MS-mediated metabolic analysis of tomato RILs was done to profile seeds and decipher the interaction between seed metabolism, environment, and genetics. This study identified several genetic regions that regulate a set of metabolites [214].

Moreover, numerous studies identified mQTLs controlling the biotic interactions in plants. With the progress in sequencing technology, more plant genomes have been sequenced with frequent use of mQTL analysis in crop plants. For example, genes for phenylpropanoid synthesis in maize [215], phenolamide regulation in maize and rice [216,217], and glucosinolate control in cabbage [218] have been identified; these byproducts are considered as defensive metabolites. The mQTL mapping pinpoints candidate genes for host–pathogen interaction and dissects the regulatory pathways that control the resistance mechanism in crop plants.

### 5.2. Metabolic Genome-Wide Association Studies (mGWASs)

The mGWAS has emerged as a powerful tool to describe the natural genetic basis of various metabolic changes in the plant metabolome. Recent studies revealed the global view of secondary plant metabolites associated with a specific trait [219]. The parallel analysis of mGWAS with phenotypic genome-wide association studies (pGWASs) in rice efficiently identified novel candidate genes that regulate variations in agronomically important traits [219]. Studies on metabolic polymorphism in rice varieties identified different kinds of flavone glycosylation and reported a positive correlation of plant growing conditions with exposure to UV-B light [220]. A total of 175 rice accessions were subjected to metabolomics-assisted GWAS analysis. About 323 associations among 89 secondary metabolites and 143 SNPs were identified, which showed two types of genetic architecture identifying secondary metabolite concentrations [221]. Dong and colleagues performed natural variation analysis and metabolic profiling of phenolamides in different rice accessions via a LC/MS-mediated targeted metabolomics technique. Spatiotemporal accumulation of many phenolamides was reported in rice cultivars. Moreover, mGWAS identified two spermidine hydroxycinnamoyl transferases, which showed natural variation in spermidine concentrations. This study revealed that gene-to-metabolic investigation via mGWAS gives a valuable technology for crop genetic improvement [217]. The mGWAS analysis has been performed to dissect biochemical and genetic variations in rice metabolism. The study identified 36 genes associated with unique metabolites that control nutritional and physiological traits. Moreover, five genes have been characterized, including three putative acyltransferases, a glucosyltransferase, and a methyltransferase [199].

The characteristics of primary and secondary metabolites can be used as metabolic markers to facilitate crop breeding for genetic improvement. In 2014, Wen and coworkers integrated the genetic expression and metabolic profiling approaches to understand the genetic diversity of various metabolites in maize kernels. In another experiment, mGWAS analysis was applied to study 289 inbred lines of maize in order to examine complex metabolic traits. About 26 metabolites were identified that are linked with SNPs and control the major objective of cinnamoyl-CoA reductase in enhancing the lignocellulosic quality in maize [222]. Similarly, mGWAS was applied to unpin the genetic pathways associated with maize kernel oil biosynthesis. A total of 368 inbred lines were used and 74 trait loci for higher oil production have been identified. These results explain the genetic basis of maize kernel oil synthesis with a possible role of speeding up marker-assisted breeding for improved oil quality and quantity [223].

Recently, metabolic profiling of winter wheat revealed the association among 18372 SNPs and identified 76 metabolites. The correlation between metabolites showed their functional association, representing the close linkage of several pathways of the citric acid cycle. The mGWAS detected a strong correlation for six metabolic features, such as sugar oligomer, L-tyrosine, pentose alcohol III, L-arginine, ornithine, and oxalic acid, among 1 and 17 SNPs. The results provide a baseline for anticipating the impact of genetic interventions on similar metabolic characters and ultimately on a particular metabolic phenotype [224]. Likewise, metabolic profiling of maize and tomato has also been reported [225,226,227]. The summary of some recent applications of mGWASs in crop improvement is presented in Table 4.

## 6. Bottlenecks Remain

In recent years, tremendous progress has been made in the field of metabolomics. However, some bottlenecks need to be adequately addressed to exploit metabolomics to its full potential. The removal of these bottlenecks will assist in exploring new platforms for crop improvement, which in turn will guarantee global food security. Current analytical tools for plant metabolomics analysis are unable to detect all the metabolites in sample tissues. This drawback is directly associated with biological modification in each cell, the complex chemical nature of metabolites in the plant metabolome, and vibrant coverage of analytical tools.

Technical bottlenecks for broad-range coverage and biological bottlenecks to draw efficient knowledge and improve understanding of whole metabolome profile are the major pitfalls in metabolomics research. Due to the wide range, multiple dynamics, and diverse chemical composition, it is hard to identify metabolites using current analytical techniques as compared to RNA sequencing. Advancements in analytical instruments may lead to precision, such as improved NMR has been exploited for a whole metabolite coverage. The coverage is expected to be progressively improved with the modernization of technology that we have at this time. This would need a holistic integrated approach, whereby several tissues from diversified species can be examined by executing numerous ways for sample extraction and evaluating them on available metabolomics stages.

Still, the identification and elucidation of a large number of unexplored plant metabolites is the major hurdle in metabolomics. The coordination of different enzymes in a metabolic process together with the complexity of metabolic pathways makes it a challenging task to change the features of a single metabolite in a biological process. Hence, these metabolic characteristics hinder direct evaluation of any specific metabolite through conventional approaches of reverse genetics. To cope with this issue, highly sophisticated approaches are needed for efficient modifications in enzymes’ kinetic features. While designing advanced tools to overcome these hurdles is expected involve more sophisticated approaches, we hope that they will be beneficial in assisting the exploration of the accurate features of any metabolite and elucidate the biological function to study a particular metabolite successfully. In the end, we would propose that deciphering the function of a specific metabolite represents the big hurdle remaining in the third decade of metabolomics.

## 7. Conclusions and Future Outlook

Metabolomics has attained a prominent place in plant biology research. It has vast applications in plant sciences, ranging from exploring various climatic stresses, probing the functions of candidate genes for analyzing the whole biological mechanism in cells, and dissecting thee genotype–phenotype relationship in response to various stresses. However, plant metabolomics still demands extensive research and proper attention for data mining, data annotation, assessment, processing, and evaluation. The integration of advanced bioinformatics tools with omics approaches proficiently dissects novel metabolic networks for crop improvement.

The integration of metabolomics with post-genomics tools and genetic approaches has offered an exciting way forward to study the genetic regulations of plants in the context to their metabolism. Metabolomics has immense potential in the field of genetic breeding. Exploitation of high-throughput genome sequencing, reverse genetics, and metabolomics tools has significantly reduced the varied development time via metabolomics-assisted breeding. The combination of omics approaches, such as genomics, transcriptomics, and metabolomics, has huge potential for exploring complex metabolic pathways that govern important regulatory processes in plant metabolism. Unblocking certain bottlenecks, like decoding the structure of metabolic networks, the effect of artificial screening on crop metabolomes, and the relationship among metabolism and phenotyping, needs more dedicated efforts.

Future applications of metabolomics may include identifying metabolic markers to study plant metabolism and to predict the nature and scale of biotic/abiotic stress. Approaches like metabolomics-assisted breeding will find wide applications in crop improvement programs to develop high-yielding, stress-tolerant germplasm and create climate-smart crop varieties. Furthermore, metabolomics approaches can be used for metabolic profiling of genome-edited plants using a modern genome-editing toolkit like the CRISPR/Cas9 system [228] for risk assessment and regulatory affairs associated with genetically engineered crops. Speed breeding is yet another fascinating area where metabolomics is ready to do wonders for crop improvement.

## Figures and Tables

**Figure 1 metabolites-09-00303-f001:**
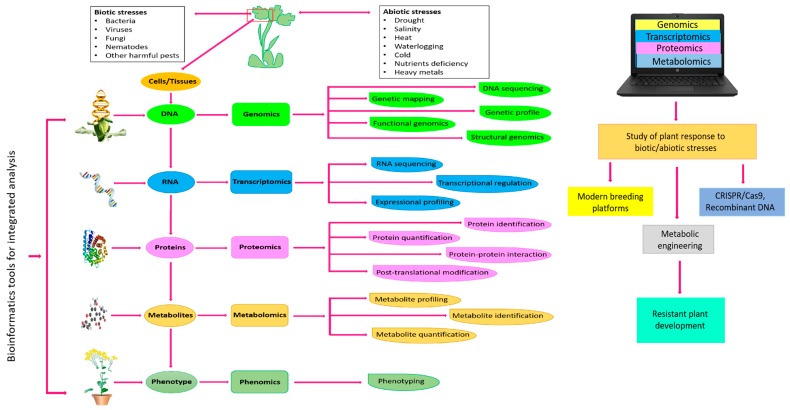
System biology for understanding the plant stress mechanism. The central dogma of plant biology showed integrated applications of genomics, transcriptomics, proteomics, metabolomics, and phenomics under biotic/abiotic stresses. Different bioinformatics tools are applied for integrated analysis to study plant stress responses from the genome to phenome levels. The data generated from these analyses are exploited for metabolic engineering and can also be executed in modern breeding platforms to generate gene edited mutants via clustered regularly interspaced short palindromic repeats/CRISPR-associated proteins (CRISPR/Cas9)/recombinant DNA technology to develop resistant crops.

**Figure 2 metabolites-09-00303-f002:**
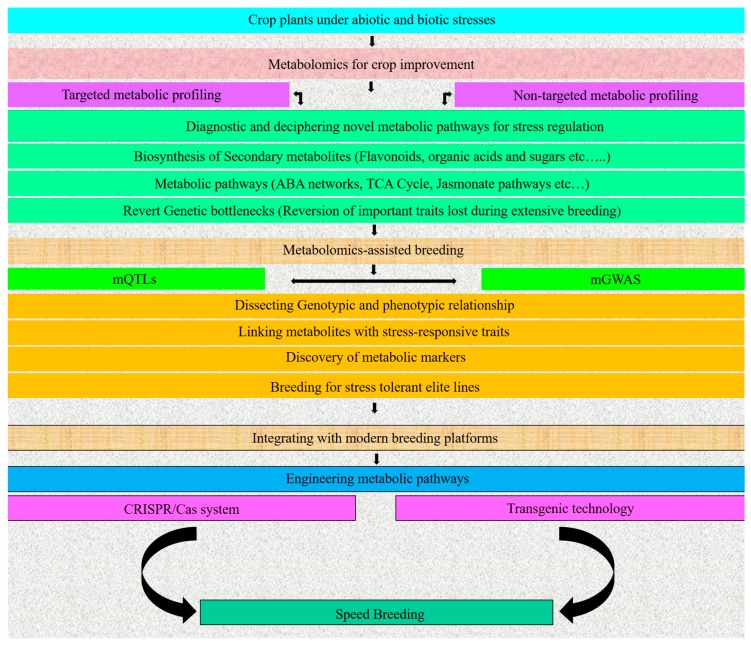
Flowchart outlining the board mechanisms in plant metabolomics for crop improvement.

**Figure 3 metabolites-09-00303-f003:**
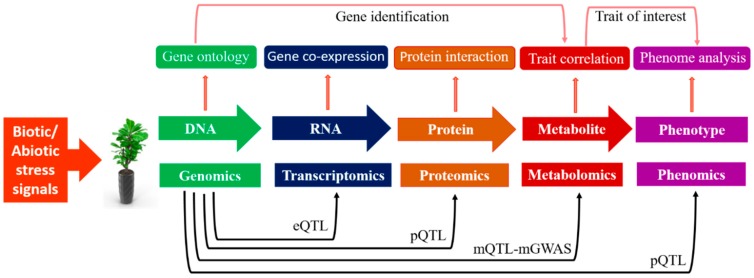
Quantitative trait loci (QTL) mapping for gene expression or a molecular phenotype. The flow of molecular information is represented from the DNA to the phenotype in response to biotic/abiotic stress signals. Black arrows indicate that each molecular phenotype can be mapped by using QTL mapping and genome-wide association studies (GWAS) techniques. Whereas, metabolic genome-wide association studies (mGWAS) does not require genetic information to investigate the effects of genetic deviations on metabolites. Red arrows show the corresponding levels of a specific gene, protein and metabolite. (eQTL: epigenomic QTL; pQTL: proteomic QTL; mQTL: metabolomic quantity trait loci; mGWAS: metabolomic genome-wide association studies).

**Table 1 metabolites-09-00303-t001:** Comparison of commonly employed tools in plant metabolomics.

Analytical Tool	Applications	Advantages	Disadvantages	Properties
Nuclear Magnetic Resonance Spectroscopy (NMR)	Non-destructive; examination of metabolites; Comparative analysis of samples	Quantitative; Highly reproducible; Accurate quantification; Robust analysis; Ease of sample preparation; Provide rich information about metabolite structure; Separation not needed; Compatible with solids and liquids	High cost of instrument; Low sensitivity; Lack of bioinformatics platform; Large volume of sample is required; Spectral analysis hectic and time-consuming	Mass range: <~50 kDa; Sensitivity: Low (10^−6^ M)
Liquid Chromatography-Mass Spectrometry (LC-MS)	Good for detection of polar compounds; Suitable for secondary metabolite analysis like vitamins, glucosinolates; flavonoids and carotenoids; Ionization method: Atmospheric pressure chemical ionization (APCI) and electrospray ionization (ESI)	High sensitivity; Good selectivity; Less volume of sample required; Derivatization not needed; Minimal sample preparation; Covers a large portion of the metabolome	Destructive; Low separation of LC column; Reduced quantification; Ion suppression; Suitable for targeted profiling; Laborious sample preparation	Mass range: <1500 Da; Accuracy: 50–100 ppm; Sensitivity: High (10^−15^ M)
Gas Chromatography-Mass Spectrometry (GC-MS)	Good for hydrophobic and polar compounds such as vitamins, organic acids, sugars, hydrocarbons and essential oils having a low molecular weight Ionization method: Electron impact (EI)	More accurate; High resolving power; Suitable for volatile compound analysis; Good sensitivity; Economical than NMR and LC-MS; Supported by bioinformatics and databases; Reproducible	Derivatization required; Destructive; Unsuitable for non-volatile compounds; Possible loss of pseudomolecular ion	Mass range: <350 Da; Accuracy: <50 ppm; Sensitivity: High (10^−12^ M)
Fourier-Transform Infrared Spectroscopy (FT-IR)	Detection of unknown metabolites analysis conducted based on mass to charge ratio (m/z) ion chemistry high-resolution MALDI	High-throughput analysis; Cost-effective; Direct characterization and separation in mixed samples; Provide more information about data	Not feasible for wet samples; Less specificity; Limited dynamic range; Isomer-related issues	Mass range: <1500 Da; Accuracy: <1 ppm; Sensitivity: High (10^−18^ M)

**Table 2 metabolites-09-00303-t002:** List of bioinformatics and statistical tools for plant metabolomics workflow.

Tool	Weblink	Major Function	Reference
MetaboAnalyst	www.metaboanalyst.ca/	Statistical analysis	[103]
MetaboSearch	http://omics.georgetown.edu/metabosearch.html	Data annotation	[104]
MeltDB 2.0	https://meltdb.cebitec.uni-bielefeld.de	Data processing	[93]
metaP-server	http://metabolomics.helmholtz-muenchen.de/metap2/	Data analysis	[105]
MetExplore	http://metexplore.toulouse.inra.fr	Pathway analysis	[106]
Metabox	https://github.com/kwanjeeraw/metabox	Analysis workflow	[100]
METLIN	https://metlin.scripps.edu/	Metabolite annotation	[89]
MetAlign	www.metalign.nl	Data processing & statistical analysis	[95]
MetiTree	http://www.metitree.nl/	Data annotation	[107]
Metab	www.metabolomics.auckland.ac.nz/index.php/downloads	Workflow analysis	[74]
MetabR	http://metabr.r-forge.r-project.org/	R package	[79]
MetaboAnalystR	https://github.com/xialab/MetaboAnalystR	R package	[80]
Lilikoi	https://github.com/lanagarmire/lilikoi	R package	[81]
MetaboDiff	http://github.com/andreasmock/MetaboDiff/a	R package	[82]
MetFrag	http://c-ruttkies.github.io/MetFrag	Metabolite annotation	[108]
MetaGeneAlyse	http://metagenealyse.mpimp-golm.mpg.de/	Metabolite data analysis	[90]
Metacrop 2.0	http://metacrop.ipk-gatersleben.de	Data annotation	[109]
MetAssign	http://mzmatch.sourceforge.net/	Data annotation	[110]
MET-COFEA	http://bioinfo.noble.org/manuscript-support/met-cofea/	Data processing	[111]
MetPA	http://metpa.metabolomics.ca	Pathway analysis	[91]
iMet-Q	http://ms.iis.sinica.edu.tw/comics/Software_iMet-Q.html	Data processing	[94]
Babelomics 5.0	http://www.babelomics.org/	Statistical analysis	[101]
XCMS	https://xcmsonline.scripps.edu	Data processing	[85]
MZedDB	http://maltese.dbs.aber.ac.uk:8888/hrmet/index.html	Data annotation	[97]
MassBank	http://www.massbank.jp/	Metabolite annotation	[112]
MaxQuant	https://www.maxquant.org/	Data annotation & processing	[113]
MetFusion	http://mgerlich.github.io/MetFusion/	Integrated compound detection	[114]
MAVEN	https://maven.apache.org/	Data processing	[115]
MZmine2	http://mzmine.github.io/	Data processing	[116]
MSEA	http://www.metaboanalyst.ca/	Pathway analysis	[92]
MS-Dial	http://prime.psc.riken.jp/Metabolomics_Software/MS-DIAL/	Data processing	[96]
MarVis	http://marvis.gobics.de/	Metabolite annotation	[117]
Mummichog	http://mummichog.org	Pathway analysis	[118]
MMCD	http://mmcd.nmrfam.wisc.edu/	Metabolite annotation	[119]
COVAIN	http://www.univie.ac.at/mosys/software.html	Statistical analysis	[62]
CAMERA	https://bioconductor.org/packages/release/bioc/html/CAMERA.html	Data annotation	[120]
CDK	https://cdk.github.io	Structural annotation	[121]
CFM-ID	http://cfmid.wishartlab.com	Metabolite identification	[122]
ADAP	http://www.du-lab.org/software.htm/	Data processing	[123]
KEGG	http://www.genome.jp/kegg/	Metabolic models	[98]
GenePattern	http://software.broadinstitute.org/cancer/software/genepattern/	Statistical analysis	[102]
Galaxy-M	https://github.com/Viant-Metabolomics/Galaxy-M	Workflow analysis	[99]

**Table 3 metabolites-09-00303-t003:** Recent applications of metabolomics platforms to decipher abiotic and biotic stress tolerance in major crop plants.

Crop	Stress Condition	Analytical Platform	Specific Tissue	Key Metabolites Produced	Data Analysis	Reference
**Abiotic Stress Tolerance**						
Maize	Drought stress	RP/UPLC-MS/MS	Immature kernels	Metabolism of lipids, carbohydrates and glutathione cycle	PLS-DA KEGG	[14]
Maize	Drought stress	GC-TOF-MS	Multiple tissues	Adenine, phenylalanine, isoleucine, succinic acid, pyruvic acid, alanine, proline and xylose	ANOVA and PCA	[141]
Maize	Drought stress	GC/MS	Leaf blades	Myoinositol and glycine	ANOVA and PCA	[4]
Barley	Drought stress	MS-EI	Fifth leaf and Palea	Aromatic amino acids, proline, glutamine, threonine, aspartate, glycine and serine	PROC UNIVARIATE, SAS v. 9.4	[16]
Wheat	Drought stress	GC-MS	Roots and leaves	Malic acid, fumaric acid, citric acid, valine and tryptophan	PLS-DA, KEEG	[17]
Wheat	Drought stress	GC/MS	Flag leaves	Glutamine, serine, methionine, lysine and asparagine	MetabolomeExpress	[20]
Wheat	Drought stress	GC-TOF-MS	Shoots	Malic acid, mannose, fructose, sucrose and proline	SIMCA 14.0, PCA, KEGG, MetaboAnalyst	[128]
Rice	Drought stress	GC-MS	Leaves	4-hydroxycinnamic acid, ferulic acid, stearic acid and xylitol	PCA, PLS-DA	[139]
Rice	Drought stress	GC/EI-TOF-MS	Leaf	Glutamate, proline, GABA, arginine and spermidine	TagFinder and NIST	[140]
Rice	Drought stress	GC/MS	Leaf blades	Serine, threonine and asparagine	PCA	[143]
Soybean	Drought Stress	H-NMR	Leaf	Glutamine, GABA, allantoin, pinitol and myoinositol	PCA	[142]
Sorghum	Drought stress	FT-IR and GC/MS	Leaf	Sugars and sugar alcohols	PC-DFA	[44]
Rice	Salt stress	GC/MS	Leaf	Mannitol and sucrose	ANOVA and MassHunter MS	[23]
Rice	Salt stress	GC-MS	Seedling	Leucine, isoleucine, valine, proline and GABA	ANOVA and DMRT	[15]
Rice	Salt stress	NMR	Leaf and root	Acetic acid, GABA, sucrose and non-polar metabolites	PLS-DA	[18]
Rice	Salt stress	GC-MS	Leaf	Vanillic acid, 4-hydroxybenzoic acid, palmitic acid, stearic acid, raffinose, L-tryptophan and pyruvic acid	PCA, PLS-DA and MetaboAnalyst 3.0	[146]
Wheat	Salt stress	GC/MS	Leaf	Proline, lysine, alanine and GABA	METABOLOMEEXPRESS	[9]
Wheat	Salt stress	HPLC	Roots and Shoots	Malic acid, proline, fructose, mannose, glycine, Glutamic acid	ANOVA, PCA,	[149]
Wheat	Salt stress	GC-TOF/MS	Leaf	Lysine, proline, sorbitol, lyxose and sucrose	PCA, OPLS-DA, KEGG and MetaboAnalyst	[144]
Maize	Salt stress	GC-MS	Leaf	Auxin, ABA	PCA, PLS-DA and SIMCA	[150]
Barley	Salt stress	GC/MS	Roots	Proline, sucrose, xylose and maltose	MetaboAnalyst	[147]
Tomato	Salt stress	UHPLC-ESI/QTOF-MS	Terminal leaflet	Sesquiterpene lactones, alkaloids and poluamines	ANNOVA, PCA, PLS-DA	[129]
Soybean	Waterlogging	CE/MS	Leaf	Phosphoenol pyruvate, NADH2, glycine and gammaaminobutyric acid	ANOVA	[22]
Soybean	Waterlogging	NMR	Roots and leaves	Isoflavones and kaempfero	ANOVA, PCA and MATLAB	[152]
Wheat	Waterlogging	GC/MS and LC/MS	Shoot	Lysine, proline, methionine and tryptophan	ANOVA and PCA	[153]
Rice	Waterlogging	GC/MS	Leaf	Glycine, alanine and GABA	PCA and MarkerLynx XS	[151]
Rice	Waterlogging	GC/MS and NMR	Leaf	6-phosphogluconate, phenylalanine and lactate	ANOVA and PCA	[154]
Wheat	Heat stress	LC-HRMS	Flag leaves	Pipecolate and L-tryptophan	PLS-DA, KEGG	[25]
Wheat	Heat stress	LC-MS/MS HPLC	Filling grains	G1p and sucrose	Metaboanalyst 2.0 and KEGG	[156]
Wheat	Heat stress	GC-MS	Leaves	Melibiose, serine, lysine, glycine, malic acid, mannitol, xylitol, inositol, fructose, proline, glutamic acid and alanine	LSD	[158]
Tomato	Heat stress	GC-MS	Fruit pericarp	Rhamnose, putrescine, myoinositol, allantoin and alanine	PCA	[155]
Tomato	Heat stress	LC-QTOF-MS	Pollens	Flavonoids	MetAlign, METLIN, PCA and ANNOVA	[159]
Soybean	Heat stress	LC-MS, GC-MS	Seed	Ferulate, naringenin-7-O-glucoside, genistein, glycitein and apigenin	PCA	[157]
Maize	Heat stress	NMR	Leaf	Sucrose, fructose, GABA, aspartate, asparagine, valine, inositol, analine and proline	PCA and SIMCA	[160]
Canola	Metal stress	NMR	Roots and leaves	Hydroxycinnamic acids and glucosinolates	PCA, ANOVA and MultiExperiment Viewer	[163]
Sunflower	Metal stress (Cr)	capHPLC-ESI(−)-QTOF-MS	Roots and leaves	Fatty acids	PLS and MetaboScape	[161]
Soybean	Metal stress (Mo)	UPLC	Roots and leaves	Citric acid, D-glucarate, gluconic, L-nicotine, and flavonoids/isoflavone	PCA, KEGG, Metlin	[43]
Wheat	Nitrogen stress	GC-MS and LC-MS	Leaf	Tyrosine, lysine, allo-inositol and L-ascorbic acid	MS-excel package	[11]
Wheat	Nitrogen stress	GC-TOF-MS	Leaf	Fucose, ribulose, lyxose, galactinol and erythritol	PCA	[166]
Wheat	Low-nitrogen stress	UPLC-QTOF	Flag leaf	Methylisoorientin-2″-O-rhamnoside, iso-orientin and iso-vitexin	PCA, OPLS-DA, Markerlynx XS™, SIMCA-P	[172]
Barley	Sulfur stress	UPLC	Roots and leaves	sulfur metabolites, organic acids and amino acids	PCA, ANOVA, MassLynx and Progenesis QI	[168]
**Biotic stress tolerance**						
Wheat	*Zymoseptoria tritici*	FT-ICR-MS	Leaf	Flavonoids, hydroxycinnamic acid amides and cinnamyl alcohols	MetaboScape 4.0, DataAnalysis 5.0 and KEGG	[26]
Wheat	*Fusarium graminearum*	NMR	Leaf	Trehalose, asparagine, phenylalanine, myoinositol, 3-hydroxybutarate and L-alanine	PCA, MestReNova 9.1.0 and Matlab	[21]
Wheat	Fusarium graminearum	NMR	Spikelet	Spermine, putrescine, GABA, inositols, galactose and lactic acid	PCA, MestReNova 9.1.0 and Matlab	[180]
Wheat	Wheat streak mosaic virus	UPLC-QTOF/MS	Leaf	Reduction in some amino acids such as L-tyrosine, tryptophan, isoleucine and phenylalanine	PCA, KEGG, METLIN, MetFrag and MetaboAnalyst	[41]
Wheat	*Fusarium graminearum*	LC-LTQ-Orbitrap	Rachis and spikelet	Fatty acids, terpenoid, phenolic glycosides, flavonoid and phenylpropanoids	MetaXCMS	[179]
Wheat	*Triticum turgidum*	LC/MS	Leaf	benzoxazinoids	PCA, XCMS and CAMERA	[188]
Rice	*Orseolia royzae*	GC/MS	Leaf	Heneicosanoic acid, threonic acid, palmitoleic acid, palmitic acid, nonadecanoic acid and linoleic acid	ANOVA	[181]
Rice	*Xanthomonas oryzae* pv. oryzae	GC/TOF and LC/TOF	Leaf	Phenylalanine and tyrosine	KEGG, MassHunter, GeneSpring-MS 1.2 and METLIN	[182]
Rice	*Magnaporthe grisea*	NMR, GC/MS and LC/MS	Leaf	Cinnamate, proline, glutamine and malate	PCA and MATLAB	[183]
Rice	*Rhizoctonia solani*	CE/TOF-MS	Leaf	Jasmonic acid, mucic acid and glyceric acid	MPP software	[42]
Rice	*Nilaparvata lugens*	GC/MS	Leaf sheath	GABA and glyoxylate	PCA and PLS-DA	[186]
Rice	*Chilo suppressalis*	UHPLC-MS and GC-MS	Leaf	Terpenoids and phenylpropanoids	KEGG	[187]
Maize	*Fusarium graminearum*	LC/MS	Roots	metabolites smiglaside and smilaside A	ANOVA and SAS software	[24]
Maize	*Bipolaris maydis*	FT-IR and NMR	Leaf	lignin, flavonoids and polyphenols	PCA	[184]
Maize	*Ostrinia furnacalis*	HPLC-MS/MS	Leaf	Phtohormones and benzoxzinoids	KEGG, PLS-DA	[185]
Tomato	*Pseudomonas syringae* pv	NMR and LC/MS	Leaf	Flavonoid and phenylpropanoids	PCA, PLS-DA	[178]
Rice	*Lolium perenne*	LC-QTOF-MS	Root and shoot extracts	3,5,6,7,8-pentahydroxy flavones, p-hydroxybenzoic acid and sinapyl alcohol	ANOVA and LSD	[189]
Wheat	Weeds	LC-MS/MS Q Trap	Root and shoot extracts	Benzooxazinoids	Analyst software	[190]
Wheat	*Lolium rigidum Urochloa panicoides*	LC-MS/MS Q Trap	Root and shoot extracts	Hydroxamic acids and Benzoxazinoids	Analyst software	[191]
Legumes	*Weeds*	UHPLC QTOF-MS	Root and shoot extracts	Flavonoids	METLIN	[192]
Wheat	Pathogen resistance	Py-FIMS	Soil rhizosphere	Glutarimide, consabatine, methylpyrrole, arachidonic acid, gibberellic acid and diacetyllycopsamine	PCA	[194]
Cereals	*Rhizoctonia solani*	LC/MS and ^1^H NMR	Soil rhizosphere	macrocarpal	PCA, PLS-DA, ANOVA and Matlab	[195]
Crop plants	*Bacillus subtilis*	NMR	Soil rhizosphere	Antimicrobial compounds	PCA	[196]

**Table 4 metabolites-09-00303-t004:** Some applications of metabolomics-assisted breeding for crop improvement.

Crop	Analytical Tool	Sample Tissue	Population	Metabolic Traits	Reference
mQTL					
Rice	LC-EI-MS	Flag leaf and seed	RILs	Metabolome	[200]
Rice	LC-Q-TOF-MS	Seed	BILs	Metabolome	[207]
Barley	LC/MS	Flag leaf	RILs	Metabolome	[208]
Barley	IC-MS, HPLC	Flag leaf	Landrace accessions	Metabolome	[209]
Maize	LC-MS	Kernel	RILs	Metabolome	[216]
Maize	LC-MS	Kernel	ILs and RILs	Metabolome	[202]
Maize	GC-TOF-MS	Kernel, leaf and seedling	RILs	Primary metabolism	[201]
Canola	HPLC	Seed and leaf	DH lines	Glucosinolates	[210]
Tomato	UPLC	Fruit	ILs	Secondary Metabolites	[211]
Tomato	UPL C-MS	Fruit	ILs	Secondary Metabolites	[212]
Tomato	GC/MS	Fruit	ILs	Metabolome	[205]
Wheat	LC-ESI-MS	Flag leaf	DH lines	Metabolome	[213]
Tomato	GC-TOF-MS	Germinating seed	RILs	Metabolome	[214]
mGWAS					
Rice	LC-E SI-MS	Grains	Landrace accessions	Metabolome	[219]
Rice	LC-QTOF-MS	Leaf	Landrace accessions	Secodary metabolites	[221]
Rice	LC/MS	Leaf	Landrace accessions	Phenolamides	[217]
Rice	LC/MS	Leaf	Landrace accessions	Metabolome	[199]
Maize	GC-MS	Leaf	ILs	Metabolome	[222]
Maize	UPLC	Kernel	ILs	Oil components	[223]
Maize	HPLC	Grain	ILs	Tocochromanol	[225]
Maize	HPLC	Grain	ILs	Carotenoid	[226]
Wheat	GC-MS	Leaf	Elite lines	Metabolome	[224]
Tomato	GC-MS	Fruit	Landrace accessions	Metabolome	[227]

Recombinant inbred lines (RILs), Inbred line (ILs), Backed-cross inbred lines (BILs), Double haploid (DH).

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
