# Peer review of "Metabolomics: A Way Forward for Crop Improvement"

_metabolites, 2019, doi:10.3390/metabo9120303_

Round 1
Reviewer 1 Report
The topic of the review is state-of-the-art and a broad review on metabolomics is still needed. In general, in the presented manuscript english language is poor and needs to be improved. As a result, it seems to me that the authors did not fully comprehend the different research topics. This is augmented by the fact that many examples are missing and the description only occurs at the surface and important details are missing.
Some topics are redundant throughout the review (the same topic is mentioned twice or several time at different places in the review). Please carefully sort the topics and remove any redundancies. Please sort your topics! Doing a table of contents for your own does help a lot here to see any redundancies and overlappings. The "mission" and the "relevance" of the review should be mentioned at the very beginning of the review. After that, analytical methods should be mentioned and than specific approaches such as the assistance with genomic approaches. Environmental (e.g. climate, desiccation) and ecological (pests, pathogens) interactions are braodly missing altogether!
You also need to be more specific. Provide specific key examples to most of the paragraphs and shorten them by the rather general description. You need to be more specific in order to catch the readers!
I would like to see the references shortened to only contain the most relevant studies. Secondly, Table 4 is like the heartstone of this review and needs to be expanded a lot by adding references to the most important studies in the particular fields. Table 4 should also be sorted by the different research disciplines like (environmental) stresses, pathogens, assisted breeding, genomics, ...
Abstract: For several years, scientists say that metabolomics is a relatively new (young) research discipline.
I think that "emerging" is enough here. Environment-gene interactions is a too narrow definition. Please broaden the scope of the review already at this point.
What is end-phenotypic?
Please shorten the introductive part of the abstract and focus more on what the review is all about. Also, a decisive conclusion would be nice.
l. 42: As many readers are not familiar with the term, please explain sustainable agriculture.
l. 44: not just biological pathways... Please describe what metabolomics is useful for (e.g. assisted breeding, robust ecotypes, stress tolerance, pathogen resistance, ...) and then narrow down the individual topics.
l. 47: It has been estimated that approx 200000 compounds exists in the plant kingdom... The vast majority is still unknown!
l. 48: please introduce the term metabolome to the readers
l. 52: interactions also exist to the environment and other organisms (please do a literature research on environmental metabolomics and ecological metabolomics and mention them shortly here)
l. 55: please describe what the difference of primary & secondary metabolites is, provide some examples and mention which analytical method can capture most of these (GC/MS, LC/MS, ...)
l. 60: please introduce the term biomarker
l. 65: "combat" is not the right word here! Please avoid altogether
l. 62-73: this paragraph should be shortened and most of the content should come first
l. 74+: You use "metabolomics tools" a lot, yet you do not describe what you are meaning by that. analytical methods like LC/MS, GC/MS, ...?
Fig. 1: Please improve the description and cite the central dogma of biology.
l. 102: please mention the 10000 discovered metabolites at l. 47.
l. 111-124: paragraph should come earlier
l. 127: now you are saying that there are 30000 metabolites deciphered which is in contrast to l. 102 !
Table 1: Please describe the technical differences of the analytical method (e.g. ionization, resolution), be more specific here!
l. 149-159: paragraph too general, please describe the technical difference between NMR and MS. NMR is a complete different technology than MS!
l. 160-180: Please focus on the technical application of MS and describe what the technical differences are and what kind of compounds can be captured.
Fig. 2: This figure is meaningless and should be removed. These kinds of figures are always incomplete and only show a fraction of possibilities. I suggest that the workflow in metabolomics should be described in the text and major other reviews that deal with the topics (e.g. statistical analyses, analytical methods) should be cited.
l. 189-224: Please provide better reference to literature, especially book chapters and other reviews. There a thousand other different methods out there... Also, standardization of methods is missing.
l. 232: metabolomics can be used with more than two plant samples and not only for stresses, you probably mean study factors, and there too, metabolomics can be used with many different factors, please broaden the scope
l. 247-255: There are more than just group-wise approaches. Please also describe the difference between univariate and multivariate methods and their use with (many different) study factors.
l. 256-276: Please be more specific. The contents of these two paragraphs could be summarized in just 2-3 sentences. What is the difference between a PLS and PCA? Please also describe the advantages and disadvantages of each method and describe for what kind of data the methods are best suited.
l. 277-289: Please add which different knowledge of the data we gain from supervised vs. unsupervised methods.
3.3: Topic (statistical) biomarker discovery is missing
l. 303: I think CXMS is not just a web interface but a package for the R programming language. Please do a better literature research.
l. 310: what do you mean by metabolic fingerprinting?
l. 320: The description is a bit outdated. Please describe how you access the different services and how you integrate the data between them. Surely there are references that are doing that.
l. 341: Why are you introducing microbe interactions here while you are mentioning stresses? Do you mean pathogens? Please also provide examples!
Fig. 3: The flow between the different compartiments is not clear to me.
l. 358-381: Please provide examples.
l. 433-446: Please shorten, only focus on the key discoveries and provide more 1-2 important examples.
Fig. 4: Figure desription missing, please also list more examples here. There a (literally) thousands of different studies on the different crops, there have to be more and better studies than listed here......
4. + 5.: incomplete, please provide better examples
l. 669-671: Not clear to me what you want to say...
l. 673: Coverage by analytical or informatic advancements?
l. 693-710: What do you mean? Please shorten paragraph.
References: you cite too many studies. First of all, I would like to see the list shortened to only the most relevant studies. Secondly, Table 4 is like the heartstone of this review and needs to be expanded a lot by adding references to the most important studies in the particular fields. Table 4 should also be sorted by the different research disciplines like (environmental) stresses, pathogens, assisted breeding, genomics, ...
Reviewer 2 Report
The review entitled "Metabolomics: A way forward for crop improvement" deals with an interesting work regarding metabolomics and agriculture and biotechnology streams.Comments:
English language can be further modified by the help of a proofreader or by English naive speaker.
References and references' list should be revised in line with the authors guideline.
Some typographic errors were detected, please revise nd modify where applicable.
Figures can be further improved
Round 2
Reviewer 1 Report
The manuscript has gained quality, but it still needs some work. English still needs to be improved.
Please make your statements less absolute. Often there are many alternatives and there is no single fit-them-all solution. See comment for line 289.
Please make the entire manuscript more concise and shorten sentences. See comment for line 32. There is redundant information in some sentences that should be cleared.
l. 32 Please shorten sentences: We summarize possible bottlenecks in plant metabolomics and outline possible future prospects.
l. 50-75 ecological met is sometimes confused with environmental met (abiotic only). needs more clarity
l. 77 why is this paragraph in bold letters?
l. 289 Please make the sentence less absolute, e.g. A Person's test _can_ be used to ... CCA is often used to ... This is also the case with some other statements throughout the article
l. 297 afaik ROC is usually being used to evaluate the goodness of a statistical model, not to test for significant effects, consider removing or providing a qualified reference
l. 306 PLS-DA not PLS,DA
l. 326 Nowadays, the statistical language "R" is used most often by scientists. Please mention R briefly here or a review that deals with the topic and the 3-4 most important packages for assisted breeding and metabolomics.
Table 3 I would still like to see the Table expanded. Please add some examplary studies of important pests (i.e. insects like Ostrinia on maize) + interfering weeds. Soil is also a very important topic. I would like to see 4-5 studies on soil and metabolomics (belowground) as well.
Reviewer 2 Report
Please consider revise the English language by professional native or proofreader
References and reference list to be revised in line with author guidelines
Round 3
Reviewer 1 Report
There are still formatting errors and the English has been improved for the worse. This should not happen in a third revision! Please go very carefully over the manuscript again. i.e. just look at the first two sentences of the abstract. They can be improved considerably! In the entire manuscript there are now missing propositions, articles (the, a) and terms are in the wrong plural or singular form.
Please also avoid complicated words that need introduction (i.e. in Abstract: what is morphometric, what is probing? first occurrence here). Many sentences can still be shortened so that they are easier to read and understand. My earlier remarks have still not been addressed in this regard!
Please do not look only for formatting errors, but also make sure that your sentences make sense! There are many sentences that are hard to understand or imprecise.
And also do not use yellow background to indicate the changes - which is very hard to read. Please use i.e. light grey instead.
Suggested changes in uppercase.
"Metabolomics an emerging branch of “OMICS” science is a set of tools exploited for
identification and quantification of metabolites or chemical footprints of cellular regulatory processes in different biological systems."
should be:
Metabolomics IS an emerging branch of “omics” AND IT INVOLVES the identification, quantification of metabolites AND chemical footprints of cellular regulatory processes in different biological SPECIES.
Second sentence of abstract:
Metabolome is the total metabolite pool in an organism, that can be employed for its morphometric probing induced by genetic or environmental variations.
should be:
THE Metabolome is the total metabolite pool in an organism, that can be MEASURED TO CHARACTERIZE genetic or environmental variations.
Abstract: "phenotype depiction", the term phenotyping is more common I think
l 35: Not a sentence: "For sustainable agricultural production which focus on the continuous yield increase and improve nutritional value of crops by adopting integrated approaches."
l 39: THE Plant kingdom containS A huge diversity of metabolites of APPROX. 200000 compounds and THE vast majority IS still unknown.
l 49: Plant growth and development under different environmental conditions ARE greatly influenced by the synthesis of A large number of metabolites
...
Figures: Please remove the brown'ish background from the figures and convert the colors to a color that is good to read. Please also provide non-rasterized versions that are not scaled.
l 195: Sample preparation is ONE OF THE most crucial partS IN metabolomics AS IT HAS A TREMENDOUS INPACT ON THE final results
please do not use the term credentials as it is used in comoputer science for security and login.
please do not use the term precision here, it is usually used in mathematics/statistics
Please avoid using "etc" and use these terms: such as, like
l 200: THE "Main objective of sample preparation is to eliminate unwanted elements and enrich desired
metabolites."
This is a wrong assumption. Unwanted elements are not removed! sample prep is used to divide metabolites according to which phase they solute in.
l 208. The process is known as quenching. Please use this term.
l 240: "so it is called the data-rich technique [59]." This term is new to me. Do you mean big data?
l 312: "XCMS is online scripts" what do you mean?
l 317: METLIN is a database not an algorithm
l 363 heading of paragraph misaligned
l 587 paragraph should get a heading
l 756: "Presently, 50% of plant metabolomes has been explored." Needs reference, considering that only a fraction of plant species are known this number is likely wrong, or I do not understand the context.
l 773: Please use verbs such as should, may, ... as it is very hard to predict what research is heading to.
Table 4. SOME applications of metabolomics-assisted breeding for crop improvement
